# Response of the Invasive Alien Plant *Duchesnea indica* (Andrews) Teschem. to Different Environmental and Competitive Settings

**DOI:** 10.3390/plants14111563

**Published:** 2025-05-22

**Authors:** Maja Kreća, Nina Šajna, Mirjana Šipek

**Affiliations:** Department of Biology, Faculty of Natural Sciences and Mathematics, University of Maribor, Koroška c. 160, SI-2000 Maribor, Slovenianina.sajna@um.si (N.Š.)

**Keywords:** non-native species, light availability, nutrient availability, intraspecific competition, interspecific competition, pot experiment, morphological traits, ecophysiological traits, *Glechoma hederacea*, *Potentilla indica*

## Abstract

Indian mock strawberry (*Duchesnea indica*, syn. *Potentilla indica*), a clonal invasive plant native to Asia, has rapidly spread in Europe, where its ecological adaptation allows it to thrive under varying environmental conditions. It is mostly found in urban habitats such as lawns, parks, and urban and peri-urban forests, where it thrives in various plant communities. It can become dominant in certain communities, indicating its competitive advantage over native plants. Due to similar habitat preferences, it often coexists with the native species *Glechoma hederacea*, with which it shares other characteristics such as clonal growth. This study investigates the effects of light, nutrients, and competition on the growth, morphology, and physiology of *D. indica*. A controlled pot experiment exposed plants to combinations of sunlight and shade, optimal and increased nutrient levels, and competitive scenarios with the native plant *G. hederacea*. The plant traits of biomass, leaf and ramet number, stolon and flower production, leaf greenness, the photosynthetic efficiency of Photosystem II, and stomatal conductance were assessed. Results revealed that light and nutrient availability significantly enhanced growth metrics. In shaded conditions, *D. indica* adapted with elongated petioles and increased specific leaf area. Competition significantly reduced growth, with *G. hederacea* outperforming *D. indica*. These findings highlight the complex interplay between abiotic and biotic factors in influencing invasive species impact, providing essential insights for ecosystem management.

## 1. Introduction

Invasive alien species are among the top five drivers of biodiversity loss [1]. Invasive alien plants (IAPs) disrupt native ecosystems by altering plant community composition, often leading to reduced species richness and diversity in invaded areas compared to uninvaded sites [2,3,4]. Many IAPs contribute to ecosystem changes by producing large amounts of easily decomposable litter, which affects decomposition processes and alters soil nutrient cycles [5]. The ecological success of IAPs is largely attributed to traits such as rapid growth, prolific reproduction, and high adaptability [6,7], enabling them to outcompete native flora and establish dominance, particularly in disturbed environments [8,9,10].

*Duchesnea indica* (Andrews) Teschem. (syn. *Potentilla indica* (Andrews) Th. Wolf), native to East and Southeast Asia [11], is an invasive alien plant that was introduced worldwide as a ground-cover ornamental plant and has since spread across Europe, North and South America, Australia, and New Zealand [11,12]. It is becoming a common species with the ability to form dense patches and outcompete native flora. We observed two sites in northeastern Slovenia where *D. indica* became dominant and outcompeted most of the native flora. One site was a forest understory previously overgrown by *Glechoma hederacea* L., and the other was a vineyard where *Fragaria vesca* L. was predominant (personal observation by MŠ). In Slovenia, Europe, *D. indica* is widespread in urban environments, thriving in lawns, meadows, forest understories, and other moist, nutrient-rich locations [13]. Additionally, it is spreading into natural habitats, a trend expected to continue with global climate change and rising temperatures [12]. However, its competitive ability against native plants under different environmental conditions remains poorly understood.

This study examines the effects of light intensity, nutrient availability, and competition on the growth and competitive strength of *D. indica*, comparing it to the ecologically similar native *G. hederacea.* Both species frequently coexist in anthropogenic habitats [7] and are model clonal plants [14,15,16,17,18] that share similar ecological requirements and life strategies (CR/CSR) [19,20,21]. They thrive in a wide range of habitats, from fully open to shaded environments with nutrient-rich soils [22,23,24], where plant growth is often restricted by disturbances and/or shading [25]. Given their frequent co-occurrence, competition between *G. hederacea* and *D. indica* is expected and might be dependent on environmental settings such as habitat.

Light and mineral nutrients are key environmental factors shaping plant growth across spatial and temporal scales. These factors influence a plant’s response to its environment, driving adaptations in growth, morphology, and physiology [7,14], as well as offspring development [15]. Understanding the ecological requirements of invasive species is crucial for predicting their spread and informing effective management strategies [26].

This study aims to (i) examine the effects of light and nutrients on the morphological and ecophysiological traits of *D. indica*; (ii) assess the impact of intra- and interspecific competition on its growth, and (iii) identify adaptive traits that enable *D. indica* to persist under varying environmental conditions.

We hypothesized that *D. indica* will exhibit optimal growth under high light and nutrient-rich conditions, while competition, particularly interspecific, will reduce its growth and physiological performance. Furthermore, we expect its morphological and ecophysiological plasticity to facilitate survival in suboptimal conditions, with its competitive ability increasing in resource-rich environments. Morphological and ecophysiological plasticity refer to an organism’s ability to alter its phenotype in response to specific environmental conditions [27]. This adaptive capacity is crucial for an organism’s survival, establishment, and potential spread, especially in novel or altered habitats. As such, phenotypic plasticity is hypothesized to play a key role in biological invasions. Several studies comparing native and invasive alien plant species suggest that invasive plants tend to exhibit greater phenotypic plasticity, allowing them to adapt more effectively [28,29]. However, a meta-analysis by Palacio-López and Gianoli (2011) [30] found that the plastic responses of plants are generally similar regardless of their invasive status. Nevertheless, increased phenotypic plasticity may enhance the likelihood of successful adaptation and establishment during the initial stages of invasion [30].

## 2. Results

### 2.1. Effect of Light and Nutrient Availability

#### 2.1.1. Morphological Traits

Light had the greatest influence on the growth and development of *D. indica*. The highest plant growth, assessed by the number of leaves and fruits (Table 1), as well as biomass (above ground and under ground; Figure 1), was observed in full sunlight and with higher nutrient supply, while the lowest values were recorded in the shade. Increased nutrient supply had minimal effect on the production of stolons and ramets, with plants in full sun producing an average of 14 ramets per plant and a maximum of 20. As expected, plants grown in shaded conditions exhibited significantly reduced growth, with a fourfold and fivefold decrease in the number of ramets and stolons, respectively, compared to those in full sun (Table 1). Leaf lengths, ranging from 1 to 3.9 cm, differed significantly between treatments (ANOVA, F = 4.9, *p* < 0.05).

Significant differences in leaf length were observed between plants growing in full sunlight with optimal and increased nutrient supply, while differences among the other treatments were not significant. Leaf widths measured from 0.7 to 3 cm, but they did not differ significantly between treatments (ANOVA, F = 1.4, *p* > 0.05).

Petiole lengths, ranging from 1 to 12.7 cm, differed significantly between all treatments (Kruskal–Wallis test, χ^2^ = 7.58, *p* < 0.0001). The longest petioles were observed in plants grown under shade, while the shortest were found in plants grown under full sunlight and increased nutrient supply.

Plants grown under full sunlight and increased nutrients had the highest biomass (Figure 1 and Figure 2). The biomass of the above ground parts differed significantly between all treatments (Kruskal–Wallis test, χ^2^ = 25.82, *p* < 0.0001) and under ground parts differed significantly between the sun and shade plants (Kruskal–Wallis test, χ^2^ = 19.16, *p* < 0.0001). Significantly lower biomass was recorded for shaded plants. Similar results were given for dry biomass (Figure 2).

In the shade, the dry mass of plants was very low, not exceeding 2 g, while plants growing in full sunlight with an optimal nutrient supply exceeded the dry mass of shaded plants by 13 times, and those exposed to an increased nutrient supply by as much as 18 times. The dry mass of *D. indica* growing in full sunlight with optimal and increased nutrients ranged between 3 and 6 g for the above ground part and between 6 and 7 g for the under ground part of the plants.

#### 2.1.2. Ecophysiological Traits

Ecophysiological traits of *D. indica* were less affected by experimental conditions than morphological traits. At the start of the experiment, 13 days after exposing *D. indica* to different treatments, *D. indica* exhibited similar SPAD values and photosystem II efficiency (F_v_/F_m_; chlorophyll fluorescence) across all treatments (ANOVA, *p* > 0.05), while shaded plants exhibited higher leaf stomatal conductance than plants grown under full sunlight.

The leaf stomatal conductance of *D. indica* differed significantly between treatments from the beginning to the end of the experiment (Kruskal–Wallis test, χ^2^ = 11.88, *p* < 0.05). The average stomatal conductance of *D. indica* on the initial measurement date ranged from 315.15 to 670.3 mmol/m^2^s. The lowest conductance was recorded in plants growing without competition in full sun with optimal nutrients (115.7 mmol/m^2^s), while the highest values were measured in plants growing without competition in the shade (858.5 mmol/m^2^s). On the final measurement date (21 July), the average stomatal conductance of *D. indica* decreased compared to the initial measurement date, ranging from 225 to 477.2 mmol/m^2^s (Figure 3A).

On 21 July, 48 days after exposing *D. indica* to different treatments, SPAD values (Kruskal–Wallis test, χ^2^ = 30.57, *p* < 0.05; Figure 3B) and the efficiency of photosystem II (F_v_/F_m_ value of dark-adapted leaves; Kruskal–Wallis test, χ^2^ = 7.19, *p* = 0.03; Figure 3C) were significantly affected by treatments. Chlorophyll content in the leaves for the full sun with optimal nutrients treatment averaged 64.14 ± 3.06 SPAD units, and for the full sun with increased nutrient supply treatment, it was 63.1 ± 4.35 SPAD units. SPAD values for the chlorophyll content of *D. indica* growing in the shade were significantly lower, averaging 50.71 ± 3.38 SPAD units (Figure 3B).

### 2.2. Effect of Competition

Competition significantly influenced both above ground and under ground growth of *D. indica*.

In shaded conditions, the dry biomass of both above ground and under ground parts was three times greater in *D. indica* plants growing under competition compared to those growing without a competitor. Significant differences were observed in fresh above ground (ANOVA, F = 35.72, *p* < 0.0001) and under ground biomass (Kruskal–Wallis, χ^2^ = 6.82, *p* = 0.009) (Figure 2).

In full sunlight with optimal nutrient supply, the fresh biomass of both above ground (ANOVA, F = 17.05, *p* < 0.001) and under ground parts (ANOVA, F = 24.19, *p* < 0.01) of *D. indica* significantly differed between plants growing alone and those under competition. Competition negatively affected the fresh and dry above ground biomass, whereas under ground biomass was higher when *D. indica* was grown in competition. The fresh above ground biomass was nearly twice as large in plants growing without a competitor than in those exposed to competition (Figure 2).

Under higher nutrient supply, statistically significant differences were found only in the fresh above ground biomass of *D. indica* (ANOVA, F = 15.64, *p* < 0.05), which was greater when the plants grew without a competitor (Figure 2).

Competition affected the number of leaves. The number of leaves was significantly lower when *D. indica* was exposed to competition (Kruskal–Wallis χ^2^ = 7.30, *p* < 0.05). However, there was no significant difference between intraspecific and interspecific competition (Dunn–Bonferroni test, *p* < 0.05) (Figure 4).

The competition did not affect the number of ramets (Kruskal–Wallis, χ^2^ = 1.61, *p* = 0.45), the number of stolons (Kruskal–Wallis χ^2^ = 1.00, *p* = 0.60), or the number of fruits (Kruskal–Wallis χ^2^ = 2.81, *p* = 0.09) (Figure 4).

Competition had no significant effect on leaf measurements (ANOVA, *p* > 0.05) or the ecophysiological performance of *D. indica* (Kruskal–Wallis test, *p* > 0.05).

### 2.3. Comparative Growth of D. indica and G. hederacea

In all three treatments, *G. hederacea* exhibited significantly higher dry biomass in both above ground (non-parametric *t*-test, *t* = −1.39, *p* < 0.05) and under ground (non-parametric *t*-test, *t* = −1.08, *p* < 0.05) plant parts compared to *D. indica*, with the greatest difference observed in the full sunlight treatment with increased nutrient supply. However, the number of leaves (*t*-test, *t* = 0.27, *p* > 0.05), ramets (*t*-test, *t* = −0.02, *p* > 0.05), and stolons (*t*-test, *t* = −0.2, *p* > 0.05) did not significantly differ between *D. indica* and *G. hederacea*.

Notably, across all treatments, *D. indica* exhibited significantly higher SPAD values than *G. hederacea* (non-parametric *t*-test, *t* = 7.43, *p* < 0.05).

## 3. Discussion

In this study, we evaluated the growth of the invasive alien plant *D. indica* under varying environmental conditions. Specifically, we examined the effects of light intensity, nutrient availability, and both intra- and interspecific competition. As expected, the highest growth, measured as the fresh and dry biomass of above ground parts, was observed in plants exposed to full sunlight without competition. Conversely, the highest fresh and dry biomass of under ground parts was found in plants exposed to full sunlight under competition. Increased nutrient availability did not significantly enhance the growth of *D. indica*, whether in the presence or absence of competitors. However, interspecific competition favored the native species *G. hederacea*, which exhibited, across all treatments, root and shoot biomass 65% and 160% higher, respectively, than that of *D. indica*. This suggests superior resource acquisition by *G. hederacea*.

To our knowledge, this is the first study to evaluate the response of *D. indica* to competition with an ecologically similar native plant species, *G. hederacea*, with which *D. indica* coexists in some habitats [7]. To control environmental variables, we conducted a pot experiment. Although this approach has limitations in directly translating results to natural conditions, it provides valuable insights into community dynamics and the factors that may influence community structure.

We suggest that the strong responsiveness of *Duchesnea indica*’s morphological traits to environmental variation, and relatively stable ecophysiological traits across the tested environmental gradient, supports its persistence across diverse habitats and plant communities [7,31], including suboptimal conditions such as deep shade in a forest understory. This strategy allows *D. indica* to survive in unfavorable environments and rapidly expand and colonize new areas when conditions become more favorable. Notably, vegetative reproductive traits were unaffected by increased nutrient availability or competition. This resilience in reproductive output highlights a key trait commonly associated with species possessing invasive potential, enabling them to reproduce vigorously through vegetative growth. Invasive species often exhibit prolific vegetative reproduction and high propagule pressure [32], contributing to their successful establishment and spread in new environments [33].

### 3.1. Light and Nutrients Determine the Growth of D. indica

*Duchesnea indica* growing in full sunlight with optimal nutrient levels allocated twice as many resources to under ground parts. Under lower nutrient availability, plants often allocate more resources to under ground growth, reflected in a larger ratio of under ground to above ground biomass. This allocation strategy enhances nutrient absorption [17]. Full sunlight significantly enhanced biomass and ramet production, consistent with studies on *D. indica* in its native range [16,17].

*Duchesnea indica* is a species that tolerates semi-shaded to shaded habitats. In our experiment, no significant differences were observed in leaf measurements of *D. indica* grown under varying light conditions, from full sunlight to only 10% sunlight, which is similar to light conditions in a deciduous temperate forest understory during summer. On the other hand, plants grown in full sunlight with increased nutrients exhibited 230% higher leaf counts and 334% higher ramet counts compared to those grown in shaded conditions. Both above ground and under ground biomass were higher in full sunlight with increased nutrient supply, with an approximate increase of 1800%.

Wang et al. [17] identified morphological and physiological traits of *D. indica* that contribute to its adaptation to varying light and nutrient conditions. These traits include longer petioles and increased chlorophyll content in older leaves as a response to shading, and reduced chlorophyll content in mature and old leaves when nutrient availability is lower. In our study, shaded plants adapted by elongating their petioles, while leaf area remained comparable across all treatments. Longer petioles allow the plant to position its leaves higher, improving light capture in shaded conditions or when in competition with other specimens or species. Horizontal stem or petiole elongation in response to shading is a well-documented example of phenotypic plasticity, enabling plants to overcome photosynthetic limitations and thereby improve fitness [7,34].

Studies have shown that *D. indica* growing in shaded conditions exhibit reduced fitness [16], and our results confirm this finding. We observed that in shaded conditions, *D. indica* had the lowest biomass, the fewest flowers, and a reduced production of fruits, stolons, and ramets.

Furthermore, our results did not show significantly better growth of *D. indica* under increased nutrient availability. However, plants grown in nutrient-rich environments did achieve maximum biomass for both above ground and under ground parts and exhibited the greatest growth of leaves, ramets, and fruits. Overall plant growth in shade remains relatively low; *D. indica* exhibited higher biomass, as well as greater numbers of leaves, ramets, and stolons—traits that can serve as surrogates for plant fitness—when grown in sunlight rather than shade. These findings indicate that *D. indica* performs better under conditions with higher resource availability.

Plants grown in full sunlight had higher chlorophyll content in their leaves compared to those grown in the shade. This contrasts with findings from most studies, which suggest that chlorophyll content typically increases in shaded conditions [35]. Chlorophyll content serves as an indirect indicator of photosynthetic capacity and stress in plants [36]. Factors such as nutrient availability, drought, light intensity, and leaf age all influence chlorophyll levels. According to Wang et al. [17], reduced chlorophyll content in mature leaves of plants grown in shade indicates non-adaptive phenotypic plasticity. If plasticity is adaptive and enhances survival or fitness, the direction of these responses should reflect this. In contrast, non-adaptive plasticity can reduce plant fitness [37]. Our results suggest that *D. indica* can tolerate shaded environments but at the same time experiences minor stress due to the light deficit. Comparing the initial and final values of photochemical efficiency in *D. indica* leaves confirmed that the plants were exposed to environmental stress over time. However, the plants maintained relatively high photochemical efficiency values; therefore, some caution is needed when interpreting this parameter.

Light availability also influenced stomatal conductance in *D. indica*. As expected, shaded plants exhibited slightly higher values throughout the experiment. Stomatal conductance reflects water availability and plant hydration status. Low conductance may indicate water stress or tissue damage, while high conductance suggests that the plants are well-hydrated and healthy.

### 3.2. Competition Enhances D. indica Under Ground Biomass

Interspecific competition with *G. hederacea* significantly decreased the above ground biomass, while the under ground biomass of *D. indica* significantly increased under full sunlight with optimal nutrient supply. In the presence of a competitor, *D. indica* allocated more resources to its roots, with root mass twice as large as above ground biomass. This efficient resource allocation suggests strong competitive adaptability that may contribute to its invasiveness. A more stressful environment where there is competition for water and nutrients drives *D. indica* to invest more resources into root development—an adaptive response to enhance resource acquisition under competitive pressure. A similar phenomenon has been observed in *Briza media* L. and *Festuca ovina* L. [38]. Biomass allocation among plant organs improves competitive ability by optimizing the uptake of limiting resources such as light, nutrients, and water [39].

The competitive species *G. hederacea* demonstrated a resource-use advantage, with significantly higher dry mass in both above ground and under ground organs. However, *D. indica* had significantly higher SPAD values, potentially indicating a competitive advantage in photosynthetic efficiency. Although *D. indica* had higher chlorophyll content than *G. hederacea,* this may be due to prioritizing biomass accumulation over chlorophyll production. Lau and Funk [40] proposed that invasive and native species compete for the same limited resources under stressful conditions, leading to similar performance outcomes due to constraints in extreme environments. A study on competition between *F. vesca* and *D. indica* found no direct competitive effects [41]. However, *D. indica* had a biomass advantage under high nitrogen levels, which disappeared under low nitrogen availability. This suggests that increased nitrogen due to anthropogenic factors may enhance *D. indica*’s invasiveness.

While *D. indica* under competition in high-nutrient, full-sunlight conditions achieved greater biomass than in other treatments, *G. hederacea* demonstrated a competitive advantage in growth. It accumulated significantly more biomass in both above ground and underground parts, particularly under higher nitrogen availability, confirming its superior resource uptake ability [7]. However, increased nutrient availability did not enhance competition in our experiment.

We confirm that competition negatively impacts *D. indica*’s growth. Notably, *D. indica* plants subjected to intraspecific competition under high nutrient availability experienced high mortality before the end of the experiment, whereas plants without competition survived. This aligns with findings from a meta-analysis by Adler et al. [42], which concluded that intraspecific competition is, on average, four to five times stronger than interspecific competition. The combination of high nutrient availability, intraspecific competition, and external abiotic factors (such as sunlight and temperature) likely contributed to the increased plant mortality.

### 3.3. Adaptive Traits Driving the Invasiveness of D. indica in Its Introduced Range

*Duchesnea indica* is a fast-growing, clonal species composed of multiple ramets connected by lateral above ground shoots. *Duchesnea indica* exhibits several adaptive traits that facilitate its establishment and thriving along gradients of light and nutrient availability. These traits contribute to *D. indica*’s success across a broad environmental gradient, from forest understory to disturbed open habitats.

In Slovenia, *D. indica* is found in a variety of habitats, typically near urban areas, but has not yet reached more remote habitats far from settlements. A key trait enabling its tolerance of varied light conditions is its ability to allocate resources according to the most limiting environmental factor. Our results suggest that under reduced light conditions, *D. indica* allocates more resources to traits that enhance survival, such as petiole length, which help improve light capture. In environments where competition for nutrients or water is present, it tends to allocate more resources to root development for better resource acquisition. In nutrient-rich environments, the plant invests in larger above ground biomass, including leaves and flowers, as discussed above.

Vegetative reproduction through long stolons allows the plant to spread rapidly across heterogeneous microhabitats. This enables *D. indica* to successfully utilize patchily distributed nutrients and light [7,43,44,45,46] and to resist or escape competition [47]. In our experiment, plants produced between one and eight stolons, with up to 24 ramets. The adaptability of this trait is evident from the significant variation in stolon production between plants grown in full sunlight and those in shade. Rapid growth is crucial in environments with high competition. *Duchesnea indica* is tolerant of disturbances such as grazing and physical damage, partly due to its vegetative reproduction and ability to regenerate from broken stolons. This allows it to rapidly recolonize disturbed areas and persist in a habitat.

Moreover, *D. indica* has a prolonged flowering period, beginning as early as May and continuing into October, with plants still producing seeds that germinate well (personal observations by MŠ). Its early flowering and rapid fruit production provide a reproductive advantage in environments with fluctuating conditions and frequent disturbances.

While drought tolerance is a common trait of invasive plants [48], *D. indica* is not typically found in arid habitats. However, it can tolerate dry, open environments such as ruderal sites. In Maribor (Slovenia), *D. indica* was found in ruderal habitats with only 6.7% soil moisture (measurements made in 2020 by MŠ), which confirms its drought tolerance, likely linked to its ability to allocate resources to root development [48].

## 4. Materials and Methods

### 4.1. Experimental Plants

#### 4.1.1. Indian Mock Strawberry (*Duchesnea indica*)

*Duchesnea indica* (Rosaceae) is a perennial herbaceous plant with vigorous vegetative growth of long stolons above ground with ramets at the nodes that root when in contact with moist soil. The dark green evergreen leaves have three leaflets and grow in rosettes. The flowering period in the introduced range lasts from May to late October. The yellow flowers are supported by a calyx and epicalyx. The upright red fruit resembles a strawberry; however, it is not palatable.

It was introduced to Europe in the early 19th century and was initially cultivated in botanical gardens before spreading beyond cultivated areas. By 1944, it was recorded as a rare plant in Vienna, but within 50 years, it had become common in many parts of the city. The increasing number of records indicates its spontaneous spread. By 2007, it was documented at 224 locations in Germany, Austria, and Switzerland and is recognized as an invasive species in several European countries, as well as in the Americas and Australia [49].

#### 4.1.2. Ground Ivy (*Glechoma hederacea*)

*Glechoma hederacea* (Lamiaceae) is a polycarpic perennial plant native to temperate regions of Eurasia, widely distributed in grasslands and forests across Europe. The plant has horizontally growing long stolons that can exceed 1 m with multiple nodes from which two petiolate leaves emerge. At the nodes, roots develop upon direct contact with a suitable substrate [46]. Its wintergreen leaves are dark green, kidney- to heart-shaped, coarsely scalloped, and bluntly toothed, situated on petioles 3–6 cm long. Leaf diameters range between 2 and 3 cm. Labiate flowers are blue-violet, covered with tiny trichomes, and grow in pairs or threes in the axils. They have four stamens protruding from the corolla tube. The flowering period lasts from March/April to June.

*Glechoma hederacea* can form extensive clonal patches, with different parts of individual clones often growing under contrasting conditions due to habitat heterogeneity, even on small spatial scales. Vegetative spread can result in single clones covering several square meters [45].

### 4.2. Pot Experiment

To assess the effects of light, nutrients, and competition on the growth and development of the invasive *D. indica*, we chose a pot experiment. Pot experiments allow plant studies under controlled conditions without the confounding influence of heterogeneous environmental factors, which are unavoidable in field measurements. The competitive plant for the experiment was the native species with similar ecological requirements, *G. hederacea*.

The experiment pots measuring 11 cm × 8 cm were filled with vermiculite, which provides ideal conditions for plant growth by regulating moisture and improving soil aeration. Rich in potassium and magnesium—essential minerals for plant growth—vermiculite enhances root development.

For the experiment, genetically identical plants (clones) were used: 60 for *D. indica* and 15 for *G. hederacea*. *Duchesnea indica* and *G. hederacea* plants were collected and planted in the vermiculite on 3 March 2022. The alien *D. indica* plants were collected in the forest Ptičji gaj in Brezje, Maribor, Slovenia. This population was monitored from 2019 when *D. indica* outcompeted *G. hederacea* and appeared for the first time at this site. The population resulted from the fast vegetative spread of *D. indica* plants growing at the nearby forest edge (personal observation by MŠ). A few adult *D. indica* plants growing in the understory in semi-shade, originating from vegetative spread during the previous year, were carefully dug out, and separated into several ramets, which were planted in vermiculite. *Glechoma hederacea* was collected from the Na Jožefu neighborhood in Slovenska Bistrica, growing in semi-shade between a flower bed and a lawn. A large plant was dug out and each ramet was divided from the mother plant and potted individually in vermiculite to ensure that all ramets were well rooted. Since all ramets were derived vegetatively from a single parent plant, they were genetically identical, ensuring uniformity in the experimental material. Observed differences in the experiment were due to varying ecological conditions rather than genotypic differences. Plants grew outdoors in the pots and were watered with tap water from the Maribor water supply until they were collected and planted in the vermiculite on March 3rd. Watering frequency depended on external conditions, occurring 1–2 times weekly, with all pots receiving equal amounts of water. At the beginning of the experiment, the plants were pruned to ensure the same size. *Glechoma hederacea* consists of three nodes with leaves and a rosette with three leaves for *D. indica*.

#### Treatments

We exposed the plants to three different environmental conditions, referred to as treatments: light and optimal nutrients (SU), shade with optimal nutrients (SH), and light with increased nutrients (SUN).

Each treatment included three competition scenarios: a control group (one *D. indica* plant per pot), interspecific competition (one *D. indica* and one *G. hederacea* plant in the same pot), and intraspecific competition (two *D. indica* plants in the same pot). Each combination was replicated five times, yielding 45 pots (3 treatments × 3 competition scenarios × 5 replications) arranged in a randomized design.

In the SU and SUN conditions, plants were exposed to natural outdoor light conditions (Table 2). On 3 June 2022, plants under treatment SH were shaded using a shading net placed over pots, which transmitted approximately 10% of photosynthetically active radiation. This setup allowed the evaluation of plant responses to varying light conditions, while maintaining consistent watering and nutrient distribution across treatments.

From 3 June 2022, the plants were watered according to their assigned treatment using water supplemented with liquid Substral Naturen BIO^®^ fertilizer. Substral is a compound mineral fertilizer containing nitrogen, phosphorus, and potassium in an NPK ratio of 6:6:7, along with essential micronutrients. Plants under treatment with optimal nutrients were watered with a prescribed amount of Substral (7 mL/1 L water), while plants under treatment with increased nutrients were watered with double the amount of fertilizer (14 mL/1 L water).

Outdoor environmental conditions during the experiment are summarized in Table 2.

### 4.3. Measurements of Plants

Morphological and ecophysiological measurements of *D. indica* and *G. hederacea* were conducted weekly from 16 June 2022 (the first measurements) to 21 July 2022 (the last measurements). The experiment ended on 14 September 2022, when all plants were carefully uprooted from the substrate. Vermiculite was removed from the roots, and the above ground and under ground biomass was assessed. This final evaluation provided comprehensive data on plant growth and biomass allocation under different experimental treatments.

Plants subjected to intraspecific competition under full sunlight and high nutrient supply experienced high mortality before the experiment concluded, preventing biomass assessment and the final measurement of morphological and ecophysiological traits.

#### 4.3.1. Measurement of Morphological Traits

During the experiment, the following morphological parameters were measured: leaf dimensions (length and width of leaves, and petiole length), growth assessment (at the end of the experiment, plant growth was evaluated based on the number of ramets, stolons, fruits, flowers, and leaves), and the fresh and dry biomass of above ground and under ground plant parts. Fresh biomass was measured on 14 September 2022 using the precision balance Exacta 1200 EB. Dry biomass was evaluated after plants were dried in paper envelopes in an oven at 60 °C for 72 h. Leaf dimensions were measured on all plants using a single leaf that appeared to have reached its final size. In *D. indica*, a leaf from the rosette was measured, while in *G. hederacea*, a leaf from the stolon was selected.

#### 4.3.2. Measurement of Ecophysiological Traits

Ecophysiological measurements were conducted on the same leaf used for morphological measurements. The following ecophysiological measurements were conducted between 10:00 AM and 3:00 PM: chlorophyll content in leaves was evaluated as greenness using the SPAD-502 PLUS meter (Konica Minolta, Chiyoda-ku, Japan), with SPAD values measured serving as a surrogate for chlorophyll concentration [51]. Chlorophyll fluorescence was assessed using the Handy PEA fluorometer (Hansatech, King’s Lynn, UK) to evaluate the photosynthetic efficiency of photosystem II (PSII). The F_v_/F_m_ ratio was obtained from dark-adapted leaves to assess plant health and photosynthetic activity, with values below 0.83 indicating stress or photoinhibition. The F_v_/F_m_ ratio is a good predictor of the quantum yield of net photosynthesis, whereby F_v_ represents variable fluorescence and F_m_ maximal fluorescence [52,53]. Stomatal conductance [54] was measured using the SC-1 Leaf Porometer (Decagon Devices, Pullman, WA, USA).

### 4.4. Statistical Analysis

Descriptive statistics (mean, minimum, maximum, and standard deviation) were used to summarize data from the pot experiment. To assess the effects of treatments and competition on the morphological and ecophysiological traits of *D. indica*, we applied ANOVA for data meeting normality and the homogeneity of variance assumptions following a post-hoc Tukey HSD test. Otherwise, the non-parametric Kruskal–Wallis test was used to assess differences among groups, followed by the Dunn–Bonferroni test. For comparisons of growth between *D. indica* and the competing plant *G. hederacea*, a *t*-test was used. All statistical analyses were conducted in R [55].

## 5. Conclusions

Our experiment confirmed that *D. indica* can tolerate a wide range of light conditions, from deep shade to full sunlight. *Duchesnea indica* exposed to full sunlight and increased nutrient availability exhibited the highest growth rates in terms of biomass accumulation. Although growth was significantly reduced under shaded conditions, plant survival remained unaffected. Morphological traits of *D. indica* responded strongly to environmental variation, whereas ecophysiological traits such as SPAD values and PSII efficiency were less plastic, indicating a relatively conservative physiological strategy. This limited physiological plasticity may support plant persistence under suboptimal conditions, allowing *D. indica* to survive until favorable conditions arise. Competition had a measurable, yet context-dependent impact: while it reduced above ground biomass under full sunlight, it unexpectedly increased under ground biomass. Importantly, vegetative reproductive traits such as ramet and stolon production were not significantly affected by competition. This resilience of reproductive output, even under competitive pressure, highlights a key trait associated with species that possess invasive potential. Overall, *D. indica* demonstrates substantial morphological plasticity that promotes establishment and spread in favorable, resource-rich environments, while its capacity to persist in shade and maintain vegetative reproduction under competition suggests a robust strategy for survival and expansion in a variety of habitats and plant communities.

## Figures and Tables

**Figure 1 plants-14-01563-f001:**
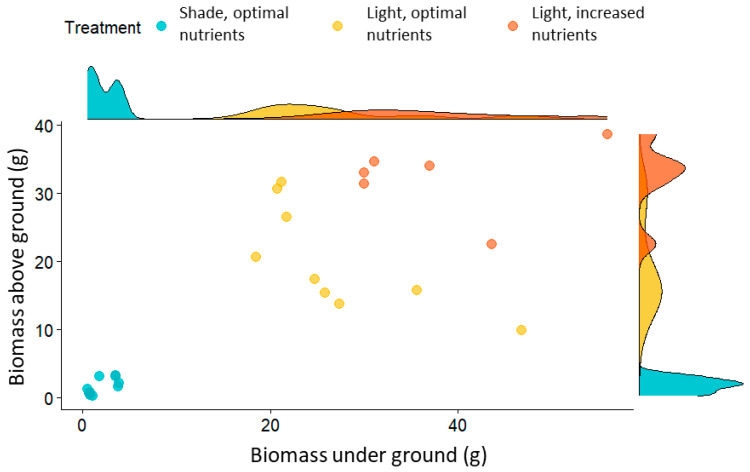
Above ground and under ground biomass of *D. indica* growing under three treatments: shade, full sunlight, and full sunlight with increased nutrient supply.

**Figure 2 plants-14-01563-f002:**
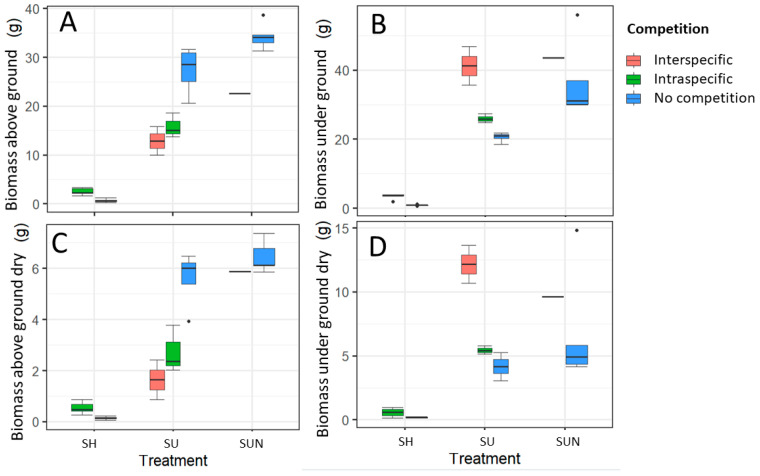
Above ground (**A**,**C**) and under ground (**B**,**D**) biomass of *D. indica* grown under three treatments—shade (SH), full sunlight (SU), and full sunlight with increased nutrient supply (SUN)—across three competition scenarios.

**Figure 3 plants-14-01563-f003:**
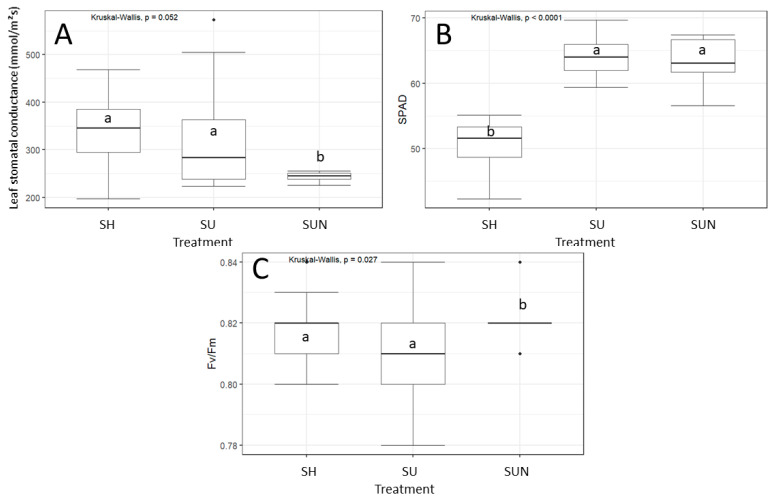
Leaf stomatal conductance (**A**), leaf greenness in SPAD units (**B**), and the efficiency of photosystem II (**C**) of *D. indica* measured on 21 July, 48 days after exposure to different treatments: shade (SH), full sunlight (SU), and full sunlight with increased nutrient supply (SUN). Significant differences between treatments are indicated by different lowercase letters.

**Figure 4 plants-14-01563-f004:**
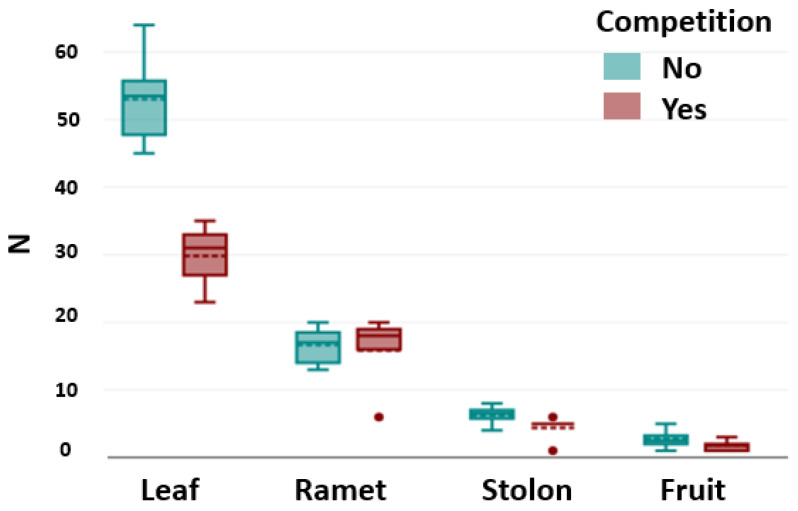
Number of leaves, stolons, ramets, and fruits of *D. indica* under control (no competition) and competition conditions (combined intra- and interspecific). Blue bars represent the control, while red bars indicate competition. Dashed lines represent the mean values, and solid lines indicate the medians.

**Table 1 plants-14-01563-t001:** Average number of leaves, ramets, fruits, and stolons of *Duchesnea indica* under different treatments with standard deviation, minimum, and maximum value. SUN—full sun with more nutrients, SU—full sun with optimal nutrients, and SH—shade with optimal nutrients. The Kruskal–Wallis (K-W) χ^2^ and *p*-values (Dunn–Bonferroni test) are shown. Significant differences between treatments are indicated by different lowercase letters.

	Treatment					
Number of		SUN	SU	SH	K-W χ^2^	* p *
** Leaves **	Average	50.17 ^a^	33.25 ^a^	15.18 ^b^	20.76	<0.05
	Std. deviation	13.92	14.28	4.64		
	Min.	24	9	9		
	Max.	64	55	23		
** Ramets **	Average	14.67 ^a^	14 ^a^	3.38 ^b^	11.82	<0.05
	Std. deviation	5.28	6.05	1.69		
	Min.	5	2	1		
	Max.	20	20	6		
** Fruits **	Average	3	2.13	1.5	1.59	>0.05
	Std. deviation	1.83	0.83	0.71		
	Min.	1	1	1		
	Max.	5	3	2		
** Stolons **	Average	4.83 ^a^	5.09 ^a^	1 ^b^	11.85	<0.05
	Std. deviation	1.72	2.12	0		
	Min.	2	1	1		
	Max.	7	8	1		

**Table 2 plants-14-01563-t002:** The average values of temperature, precipitation, and other weather parameters were recorded in Maribor during the experiment (March–August 2022). These data were sourced from the Slovenian Environment Agency’s Meteorological Service [50].

Month	Average T [°C]	Average Max T [°C]	Average Min T [°C]	Precipitation [mm]	Number of Sunshine Hours [h]	Average Cloud Cover [%]	Number of Days with Storms	Number of Days with Precipitation i > 0.1 mm
March	4.4	12.1	−2.5	5.6	229.4	37	0	1
April	9.5	15.2	3.9	87.7	213.7	63	1	12
May	17.3	23.1	11.2	69.2	236.3	61	4	13
June	21.8	27.2	15.6	73.3	296.7	50	11	12
July	22.6	29.2	15.5	53.7	312	45	4	10
August	21.7	28.4	16.2	78.8	240.5	55	6	15

## Data Availability

Data are available from the corresponding author upon reasonable request. The data are not publicly available to protect the integrity of future analyses that are currently underway.

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
