# Peer review of "Response of the Invasive Alien Plant Duchesnea indica (Andrews) Teschem. to Different Environmental and Competitive Settings"

_plants, 2025, doi:10.3390/plants14111563_

Round 1

Reviewer 1 Report

Comments and Suggestions for Authors

This study investigates the effects of light, nutrients, and competition on the growth, morphology, and physiology of D. indica, and comparing it to the ecologically similar native Glechoma hederacea L., with which it frequently coexists in anthropogenic habitats.

Abstract

The abstract emphasizes the complex interaction between abiotic and biotic factors in the impact of invasive species, but this point is not elaborated in detail in the article.

Results

The results did not examine the interactive effects of light, nutrients, and competition through statistical models, but only presented the main effects of each factor individually.

Materials and Methods

The Materials and Methods should be placed before the Results.

In the experiment, only vermiculite was used as the substrate, without considering the effects of real soil (such as organic matter content and microbial communities) on plant growth.

Line 388-391, how many pots in each control group, interspecific competition group and intraspecific competition group? The experimental design needs to be more detailed.

Author Response

We sincerely thank the reviewers for their thoughtful and constructive comments, which greatly improved the quality and clarity of our manuscript. We appreciate the time and effort dedicated to reviewing our work.

Reviewer 1:

This study investigates the effects of light, nutrients, and competition on the growth, morphology, and physiology of D. indica, and comparing it to the ecologically similar native Glechoma hederacea L., with which it frequently coexists in anthropogenic habitats.

Abstract

The abstract emphasizes the complex interaction between abiotic and biotic factors in the impact of invasive species, but this point is not elaborated in detail in the article.

Response: We agree, therefore we deleted this sentence in the abstract and added: »It is mostly found in urban habitats such as lawns, parks, and in urban and periurban forests, where it thrives in various plant communities. It can become dominant in certain communities, indicating its competitive advantage over native plants. Due to similar habitat preferences, it often coexists with the native species Glechoma hederacea, with which it shares other characteristics such as clonal growth«

Results

The results did not examine the interactive effects of light, nutrients, and competition through statistical models, but only presented the main effects of each factor individually.

Response: We decided to focus on the main effects of light, nutrients, and competition individually, rather than examining their interactive effects, because of the limited number of replicates (5 replicates), which can lead to overfitting. A limited amount of data can make it challenging to to estimate the variability and detect subtle interactions between factors because interactions often require more data to model accurately.

Materials and Methods

The Materials and Methods should be placed before the Results.

Response: The journal »Plants« requires this section after the Results.

In the experiment, only vermiculite was used as the substrate, without considering the effects of real soil (such as organic matter content and microbial communities) on plant growth.

Response: We intentionally used vermiculite to avoid the complex effects associated with real soil. We are aware that pot experiments are simplified compared to natural conditions. However, this approach allows us to better assess the effects of specific factors, which can be difficult to evaluate in natural settings due to the complexity of interactions among numerous variables.

Line 388-391, how many pots in each control group, interspecific competition group and intraspecific competition group? The experimental design needs to be more detailed.

Response: Thank you for pointing this out. We improved the text: »Each treatment included three competition scenarios: a control group (one D. indica plant per pot), interspecific competition (one D. indica and one G. hederacea plant in the same pot) and intraspecific competition (two D. indica plants in the same pot). Each combination was replicated five times, yielding 45 pots (3 treatments × 3 competition scenarios × 5 replications) arranged in a randomised design.

Reviewer 2 Report

Comments and Suggestions for Authors

This study aims to (i) examine the effects of light and nutrients on the morphological and ecophysiological traits of Duchesnea indica; (ii) assess the impact of intra- and interspecific competition on its growth, and (iii) identify adaptive traits that enable D. indica to persist under varying environmental conditions. The current research tests the statement that Duchesnea indica will exhibit optimal growth under high-light, nutrient-rich conditions, while competition — particularly interspecific — will reduce its growth and physiological performance and aims to test if morphological and ecophysiological plasticity will facilitate survival in suboptimal conditions, with its competitive ability increasing in resource-rich environments. The authors use pot experiments to test the statements applying the following treatments treatments: light and optimal nutrients (SU), shade with optimal nutrients (SH), and light 386 with increased nutrients (SUN). Also, they compare the responses to the treatment of Duchesnea indica with those of Glechoma hederacea. To assess the effects of treatments and competition on the morphological and ecophysiological traits of D. indica the authors apply the following statistical tests - ANOVA for data meeting normality and homogeneity of variance assumptions following post hoc Tukey HSD test. I find the experimental design appropriate and it makes the results and interpretations convincing. The tables are well presented and the illustrations are good. They all are well cited in the text.

Minor comments

“external sepals” is not a correct term.

The synonym of Duchesnea indica is Potentilla indica, please give his name in the Introduction so that more readers can find the paper with such engines if they use these key words.

Line 46-48

Please edit this text for clarity – “This study examines the effects of light intensity, nutrient availability, and competition on the growth and competitive strength of D. indica, comparing it to the ecologically similar native Glechoma hederacea L., with which it frequently coexists in anthropogenic habitats [7]. It sounds like copy past for the abstract of ref. [7] while obviously it uses ref [7] to motivate the experimental design.

Author Response

We sincerely thank the reviewers for their thoughtful and constructive comments, which greatly improved the quality and clarity of our manuscript. We appreciate the time and effort dedicated to reviewing our work.

The Authors

This study aims to (i) examine the effects of light and nutrients on the morphological and ecophysiological traits of Duchesnea indica; (ii) assess the impact of intra- and interspecific competition on its growth, and (iii) identify adaptive traits that enable D. indica to persist under varying environmental conditions. The current research tests the statement that Duchesnea indica will exhibit optimal growth under high-light, nutrient-rich conditions, while competition — particularly interspecific — will reduce its growth and physiological performance and aims to test if morphological and ecophysiological plasticity will facilitate survival in suboptimal conditions, with its competitive ability increasing in resource-rich environments. The authors use pot experiments to test the statements applying the following treatments treatments: light and optimal nutrients (SU), shade with optimal nutrients (SH), and light 386 with increased nutrients (SUN). Also, they compare the responses to the treatment of Duchesnea indica with those of Glechoma hederacea. To assess the effects of treatments and competition on the morphological and ecophysiological traits of D. indica the authors apply the following statistical tests - ANOVA for data meeting normality and homogeneity of variance assumptions following post hoc Tukey HSD test. I find the experimental design appropriate and it makes the results and interpretations convincing. The tables are well presented and the illustrations are good. They all are well cited in the text.

Minor comments

“external sepals” is not a correct term.

Response: Thank you for this comment. We corrected to calyx and epicalyx.

The synonym of Duchesnea indica is Potentilla indica, please give his name in the Introduction so that more readers can find the paper with such engines if they use these key words.

Response: Thank you, we added a synonym into the introduction as well as in the abstract and among keywords.

Line 46-48

Please edit this text for clarity – “This study examines the effects of light intensity, nutrient availability, and competition on the growth and competitive strength of D. indica, comparing it to the ecologically similar native Glechoma hederacea L., with which it frequently coexists in anthropogenic habitats [7]. It sounds like copy past for the abstract of ref. [7] while obviously it uses ref [7] to motivate the experimental design.

Response: We corrected the statement. The reference supports the documented coexistence of D. indica nad G. hederacea.

Reviewer 3 Report

Comments and Suggestions for Authors

Review of the article: "Response of the invasive alien plant Duchesnea indica (Andrews) Teschem. to different environmental and competitive settings"

The article addresses an important topic, namely the ecological responses of the invasive species Duchesnea indica to varying environmental and competitive conditions. However, the manuscript, in its current form, requires substantial revision to reach the standards expected for scientific publication.

Introduction

The introduction does not sufficiently introduce the topic or provide a strong foundation for the study. The authors should cite more relevant literature, especially to support key statements. For example, at the end of the first paragraph (line 35), it would be appropriate to reference studies confirming that invasive species often outcompete native flora in disturbed ecosystems (e.g.: Vilà M, Ibáñez I. 2011. Plant invasions in the landscape. Landscape Ecology  26: 461–472. https://doi.org/10.1007/s10980-011-9585-3; Rewicz A, MyÅ›liwy M, Adamowski W, PodlasiÅ„ski M, Bomanowska A. 2020. Seed morphology and sculpture of invasive Impatiens capensis Meerb. from different habitats. PeerJ 8:e10156 https://doi.org/10.7717/peerj.10156; Liendo D, Biurrun I, Campos JA, Herrera M, Loidi J, García-Mijangos I. 2013. Invasion patterns in riparian habitats: the role of anthropogenic pressure in temperate streams. Plant Biosystems 149: 289–297. https://doi.org/10.1080/11263504.2013.822434). Similarly, additional citations are needed to back up the claim that D. indica outcompetes native species (line 40) or, alternatively, information on personal observation. Moreover, reference is needed to the statement that "understanding the ecological requirements of invasive species is crucial for predicting their spread and informing effective management strategies" (lines 58–59; e.g. Pyšek, P., Jarošík, V., Hulme, P.E., Pergl, J., Hejda, M., Schaffner, U. and Vilà, M. 2012. A global assessment of invasive plant impacts on resident species, communities and ecosystems: the interaction of impact measures, invading species' traits and environment. Glob Change Biol, 18: 1725-1737. https://doi.org/10.1111/j.1365-2486.2011.02636.x).

Furthermore, the concept of morphological and ecophysiological plasticity, introduced in the hypotheses (lines 66–67), should be properly explained. The authors should elaborate on how plasticity is typically studied, what is already known about plasticity in invasive plants, and why it is a key factor in invasion success (all with references to relevant literature).

Results

The Results section contains several issues that must be addressed:

-Tables and figures are incorrectly cited. Specifically:

Lines 73 and 90 should reference Table 1, not Table 2.

Line 99 should reference only Figure 1 (or 1 and 3).

Line 103 should reference Figure 3, not Figure 2.

Lines 162–168 should include a citation to Figure 4.

The same applies to the Materials and Methods section – Line 392 should reference Table 2, not Table 1.

-There are unnecessary repetitions (e.g., the same information in lines 72–73 and 85–86).

-There are factual errors, such as stating that the highest number of stolons was observed in the SUN treatment (lines 73 and 85), whereas Table 1 clearly shows that the highest number of stolons occurred in the SU treatment.

-Contradictions exist regarding the effects on ecophysiological traits. Lines 115–117 suggest that 13 days after start of the experiment, studied traits were similar across all treatments, yet later (lines 128 and 134–135), the text states that SPAD values and PSII efficiency were significantly influenced by treatments. It is probably a question of a different timing of the measurements, but this is not clarified.

-The caption for Figure 4 must clarify the type of competition shown — whether it is intraspecific, interspecific, or a combination.

Additionally, subsection 2.3 ("Comparative growth of D. indica and G. hederacea") is very brief and lacks detail. The Methods section does not explain whether morphological and ecophysiological measurements were taken for G. hederacea as well.

Materials and Methods

There are several critical gaps and ambiguities:

  1. Why were D. indica and G. hederacea collected from different sites? If unavoidable, the habitat conditions of both collection sites should be described.
  2. It is unclear whether genetically identical clones were used for both species. The collection description for D. indica ("few plants") does not confirm clonal identity, unlike for G. hederacea. This should be verified and clarified.
  3. There is confusion regarding for how long plants in pots were watering before experiment starts: plants were reportedly watered “until March 3rd” (line 378), but plants were collected “on March 3th” (line 367).
  4. Measurement frequency is not specified (daily, weekly?) in section 4.3.
  5. Sections 4.3.1 and 4.3.2 are unclear about whether measurements were made only on D. indica or both species.
  6. Section 4.3.2 lacks important methodological details: when were measurements performed during the experiment? How many replicates were used? Were all plants of both species measured?
  7. Methodological references are missing — the authors should cite previous studies that used similar ecophysiological measurements.
  8. Abbreviations such as Fv/Fm need to be clearly explained (the same applies to figure captions, e.g. SPAD and PSII in Figure 2), and the source for the stress threshold (values below 0.83) must be cited.

Discussion

The Discussion section requires significant improvement:

-There is no convincing argument that the methods chosen will provide new insights into the invasiveness of D. indica.

-In subsection 3.1, the authors discuss plant responses to low nutrient availability, although their experiment only tested "optimal" and "nutrient-rich" conditions.

- The observed result that plants grown in full sun had higher chlorophyll content contrasts with common findings in which shaded plants usually show higher chlorophyll levels. The authors gave the explanation that D. indica can tolerate shade, but is stressed by light deficiency (lines 242-244). In the next paragraph, however, there is information that may seem a bit contradictory - high stomatal conductance in shaded plants is a sign of good hydration and health (lines 247-251). Whether the plants studied are both stressed and healthy - this should be discussed more clearly.

-Again, references to environments with limited nutrients or water (lines 301–302) are misplaced because such conditions were not tested in the experiment.

Conclusions

Additionally, the manuscript would benefit from the inclusion of a dedicated Conclusions section. At present, the key findings and their broader significance are not clearly highlighted, which weakens the overall impact of the study. A concise but well-structured Conclusions section would allow the authors to clearly summarize the main results, emphasize how the study contributes to the understanding of invasive species’ ecological strategies, and articulate how their findings could inform future research or management practices. Without such a section, it is difficult for readers to fully appreciate the scientific relevance and practical implications of the work.

In summary, the manuscript requires major revisions before it can be considered for publication. The Methods section must be clarified and expanded, the Introduction should include more literature and better theoretical framing, and the Discussion should directly address the findings and their implications for the study of biological invasions. The indicated inconsistencies, contradictions, and mistakes in the Results must be corrected, and the interpretations strengthened with more robust scientific reasoning.

Author Response

Reviewer 3:

We sincerely thank you for thoughtful and constructive comments, which greatly improved the quality and clarity of our manuscript. We appreciate the time and effort dedicated to reviewing our work.

The authors

Review of the article: "Response of the invasive alien plant Duchesnea indica (Andrews) Teschem. to different environmental and competitive settings"

The article addresses an important topic, namely the ecological responses of the invasive species Duchesnea indica to varying environmental and competitive conditions. However, the manuscript, in its current form, requires substantial revision to reach the standards expected for scientific publication.

Introduction

The introduction does not sufficiently introduce the topic or provide a strong foundation for the study. The authors should cite more relevant literature, especially to support key statements. For example, at the end of the first paragraph (line 35), it would be appropriate to reference studies confirming that invasive species often outcompete native flora in disturbed ecosystems (e.g.: Vilà M, Ibáñez I. 2011. Plant invasions in the landscape. Landscape Ecology  26: 461–472. https://doi.org/10.1007/s10980-011-9585-3; Rewicz A, MyÅ›liwy M, Adamowski W, PodlasiÅ„ski M, Bomanowska A. 2020. Seed morphology and sculpture of invasive Impatiens capensis Meerb. from different habitats. PeerJ 8:e10156 https://doi.org/10.7717/peerj.10156; Liendo D, Biurrun I, Campos JA, Herrera M, Loidi J, García-Mijangos I. 2013. Invasion patterns in riparian habitats: the role of anthropogenic pressure in temperate streams. Plant Biosystems 149: 289–297. https://doi.org/10.1080/11263504.2013.822434).

Response: Thank you for the suggestion. We added all recommended references.

Similarly, additional citations are needed to back up the claim that D. indica outcompetes native species (line 40) or, alternatively, information on personal observation.

Response: We added information on the personal observation: »We observed two sites in northeastern Slovenia where D. indica became dominant and outcompeted most of the native flora. One site was a forest understory previously overgrown by G. hederacea, and the other was a vineyard where Fragaria vesca L. was predominant.«

Moreover, reference is needed to the statement that "understanding the ecological requirements of invasive species is crucial for predicting their spread and informing effective management strategies" (lines 58–59; e.g. Pyšek, P., Jarošík, V., Hulme, P.E., Pergl, J., Hejda, M., Schaffner, U. and Vilà, M. 2012. A global assessment of invasive plant impacts on resident species, communities and ecosystems: the interaction of impact measures, invading species' traits and environment. Glob Change Biol, 18: 1725-1737. https://doi.org/10.1111/j.1365-2486.2011.02636.x).

Response: Thank you for the suggestion. We added recommended reference.

Furthermore, the concept of morphological and ecophysiological plasticity, introduced in the hypotheses (lines 66–67), should be properly explained. The authors should elaborate on how plasticity is typically studied, what is already known about plasticity in invasive plants, and why it is a key factor in invasion success (all with references to relevant literature).

Response: We added explanation with relevant refrences (L. 79 – 90).

Results

The Results section contains several issues that must be addressed:

-Tables and figures are incorrectly cited. Specifically:

Lines 73 and 90 should reference Table 1, not Table 2. Corrected.

Line 99 should reference only Figure 1 (or 1 and 3). Corrected.

Line 103 should reference Figure 3, not Figure 2. Corrected.

Lines 162–168 should include a citation to Figure 4. Added.

The same applies to the Materials and Methods section – Line 392 should reference Table 2, not Table 1. Corrected.

-There are unnecessary repetitions (e.g., the same information in lines 72–73 and 85–86). Corrected.

-There are factual errors, such as stating that the highest number of stolons was observed in the SUN treatment (lines 73 and 85), whereas Table 1 clearly shows that the highest number of stolons occurred in the SU treatment. Corrected.

-Contradictions exist regarding the effects on ecophysiological traits. Lines 115–117 suggest that 13 days after start of the experiment, studied traits were similar across all treatments, yet later (lines 128 and 134–135), the text states that SPAD values and PSII efficiency were significantly influenced by treatments. It is probably a question of a different timing of the measurements, but this is not clarified.

Response: We corrected and included the date and time, expressed in days after exposing D. indica to the different treatments, in the text.

.

-The caption for Figure 4 must clarify the type of competition shown — whether it is intraspecific, interspecific, or a combination.

Response: We corrected and included this information in the caption.

Additionally, subsection 2.3 ("Comparative growth of D. indica and G. hederacea") is very brief and lacks detail. The Methods section does not explain whether morphological and ecophysiological measurements were taken for G. hederacea as well.

Response: We corrected and added in the Methods: Morphological and ecophysiological measurements of D. indica and G. hederacea were conducted from June 16, 2022 (the first measurements), to July 21, 2022 (the last measurements).

Materials and Methods

There are several critical gaps and ambiguities:

  1. Why were D. indica and G. hederacea collected from different sites? If unavoidable, the habitat conditions of both collection sites should be described.

Response: We selected collection sites that were well known to us from previous research, having been used in earlier studies (Šipek et al. 2021, Šipek 2023: Morphological plasticity and ecophysiological response of ground ivy (Glechoma hederacea, Lamiaceae) in contrasting natural habitats within its native range. Plant Biosystems - An International Journal Dealing with All Aspects of Plant Biology, 155(1), 136–147. https://doi.org/10.1080/11263504.2020.1727981 and Ground-ivy (Glechoma hederacea L., Lamiaceae) habitats in NE Slovenia: floristic, chorological and syntaxonomic diversity) https://doi.org/10.32779/gf.6.2-3.5 )

  1. It is unclear whether genetically identical clones were used for both species. The collection description for D. indica ("few plants") does not confirm clonal identity, unlike for G. hederacea. This should be verified and clarified.

Response: Details regarding the genetic uniformity of the ramets and the sampling procedures are better explained in the Methods.

  1. There is confusion regarding for how long plants in pots were watering before experiment starts: plants were reportedly watered “until March 3rd” (line 378), but plants were collected “on March 3th” (line 367).

Response: We try to better clarify in the text. Plants were watered until they were collected.

  1. Measurement frequency is not specified (daily, weekly?) in section 4.3.

Response: We added »weekly«.

  1. Sections 4.3.1 and 4.3.2 are unclear about whether measurements were made only on D. indica or both species.

Response: We corrected so that it is clear that measurements were made on both species.

  1. Section 4.3.2 lacks important methodological details: when were measurements performed during the experiment? How many replicates were used? Were all plants of both species measured?

Response: We added missing details.

  1. Methodological references are missing — the authors should cite previous studies that used similar ecophysiological measurements.

Response: We added references.

  1. Abbreviations such as Fv/Fm need to be clearly explained (the same applies to figure captions, e.g. SPAD and PSII in Figure 2), and the source for the stress threshold (values below 0.83) must be cited.

Response: We added explanations and references.

Discussion

The Discussion section requires significant improvement:

-There is no convincing argument that the methods chosen will provide new insights into the invasiveness of D. indica.

Response: We added explanations and references.

-In subsection 3.1, the authors discuss plant responses to low nutrient availability, although their experiment only tested "optimal" and "nutrient-rich" conditions.

Response: We corrected to »lower nutrient availability«.

- The observed result that plants grown in full sun had higher chlorophyll content contrasts with common findings in which shaded plants usually show higher chlorophyll levels. The authors gave the explanation that D. indica can tolerate shade, but is stressed by light deficiency (lines 242-244). In the next paragraph, however, there is information that may seem a bit contradictory - high stomatal conductance in shaded plants is a sign of good hydration and health (lines 247-251). Whether the plants studied are both stressed and healthy - this should be discussed more clearly.

Response: We corrected.

-Again, references to environments with limited nutrients or water (lines 301–302) are misplaced because such conditions were not tested in the experiment.

Response: It is difficult to find studies with directly comparable nutrient conditions, as experimental designs and substrate types vary considerably among studies. Therefore, we chose to retain these references while acknowledging this limitation

Conclusions

Additionally, the manuscript would benefit from the inclusion of a dedicated Conclusions section. At present, the key findings and their broader significance are not clearly highlighted, which weakens the overall impact of the study. A concise but well-structured Conclusions section would allow the authors to clearly summarize the main results, emphasize how the study contributes to the understanding of invasive species’ ecological strategies, and articulate how their findings could inform future research or management practices. Without such a section, it is difficult for readers to fully appreciate the scientific relevance and practical implications of the work.

Response: Thank you for suggestion. We added conclusions.

In summary, the manuscript requires major revisions before it can be considered for publication. The Methods section must be clarified and expanded, the Introduction should include more literature and better theoretical framing, and the Discussion should directly address the findings and their implications for the study of biological invasions. The indicated inconsistencies, contradictions, and mistakes in the Results must be corrected, and the interpretations strengthened with more robust scientific re

Round 2

Reviewer 3 Report

Comments and Suggestions for Authors

The comments made in my first review have been taken into account by the Authors and the indicated changes and improvements have been implemented. The manuscript can be published in its current version.